# Isolation of PCV3 from Perinatal and Reproductive Cases of PCV3-Associated Disease and In Vivo Characterization of PCV3 Replication in CD/CD Growing Pigs

**DOI:** 10.3390/v12020219

**Published:** 2020-02-16

**Authors:** Juan Mora-Díaz, Pablo Piñeyro, Huigang Shen, Kent Schwartz, Fabio Vannucci, Ganwu Li, Bailey Arruda, Luis Giménez-Lirola

**Affiliations:** 1Department of Veterinary Diagnostic and Production Animal Medicine, College of Veterinary Medicine, Iowa State University, Ames, IA 50011, USA; juanmora@iastate.edu (J.M.-D.); pablop@iastate.edu (P.P.); hgshen@iastate.edu (H.S.); kschwart@iastate.edu (K.S.); liganwu@iastate.edu (G.L.); wilberts@iastate.edu (B.A.); 2Veterinary Diagnostic Laboratory, University of Minnesota, 1333 Gortner Ave, St Paul, MN 55108, USA; vannu008@umn.edu

**Keywords:** Porcine circovirus 3, isolation, characterization, reproductive failure, swine

## Abstract

Porcine circovirus 3 (PCV3) has been identified as a putative swine pathogen with a subset of infections resulting in stillborn and mummified fetuses, encephalitis and myocarditis in perinatal, and periarteritis in growing pigs. Three PCV3 isolates were isolated from weak-born piglets or elevated stillborn and mummified fetuses. Full-length genome sequences from different passages and isolates (PCV3a1 ISU27734, PCV3a2 ISU58312, PCV3c ISU44806) were determined using metagenomics sequencing. Virus production in cell culture was confirmed by qPCR, IFA, and in situ hybridization. In vivo replication of PCV3 was also demonstrated in CD/CD pigs (*n* = 8) under experimental conditions. Viremia, first detected at 7 dpi, was detected in all pigs by 28 dpi. IgM antibody response was detected between 7–14 dpi in 5/8 PCV3-inoculated pigs but no IgG seroconversion was detected throughout the study. Pigs presented histological lesion consistent with multi systemic inflammation characterized by myocarditis and systemic perivasculitis. Viral replication was confirmed in all tissues by in situ hybridization. Clinically, all animals were unremarkable throughout the study. Although the clinical relevance of PCV3 remains under debate, this is the first isolation of PCV3 from perinatal and reproductive cases of PCV3-associated disease and in vivo characterization of PCV3 infection in a CD/CD pig model.

## 1. Introduction

Porcine circoviruses are non-enveloped small viruses with single-stranded circular DNA genomes (~1.7–2.0 kb) within genus *Circovirus* in the family *Circoviridae* [1]. Until recently, two types of circovirus have been described in pigs. Porcine circovirus type 1 (PCV1) was originally discovered as a contaminant of pig kidney cell line (PK-15) and is nonpathogenic for pigs [2,3]. Porcine circovirus type 2 (PCV2) is the causative agent of porcine circovirus-associated diseases (PCVAD) [4], including respiratory and enteric disease, reproductive failure, porcine dermatitis and nephropathy syndrome (PDNS), and PCV2-systemic disease. PCVAD causes significant economic losses to the pig industry worldwide [5,6].

The newly discovered porcine circovirus 3 (PCV3) was first detected in 2015 in a U.S. sow farm (North Carolina) experiencing PDNS and reproductive failure [7]. Subsequently, PCV3 was detected in cases of cardiac and multi-systemic inflammation in Missouri, Minnesota and South Dakota [8]. The virus has since been identified in several countries including Poland [9], South Korea [10], United Kingdom [11], Italy [12], Thailand [13], Brazil [14], Sweden [15], Japan [16], Russia [17], Spain [18], Denmark [18], Germany [19], Colombia [20], Hungary [21], and Serbia [22]. Although recently discovered, retrospective studies indicate that PCV3 has likely been circulating in swine populations for decades prior to the initial reports [15,23,24,25].

PCV3 has a genome of 2000 nt with two major open reading frames (ORFs), ORF1 encodes the proteins involved in viral replication (Rep), while ORF2 encodes the capsid protein (Cap) and has the opposite orientation from Rep [7,8]. Phylogenetic analysis of contemporary and retrospective samples has shown a consistent mutation in certain amino acids of the cap protein. Accordingly, it has been proposed that mutations in amino acids 24 and 27 within the cap protein could be potential molecular markers to classify PCV3 into three clades: PCV3a, PCV3b and PCV3c [26]. PCV3a has been further divide into three subclades (PCV3a1, PCV3a2 and PCV3a3) based on the evolutionary relationship and other molecular features of the cap protein [26].

The rapid development of PCV3-specific diagnostic detection methods has enabled the detection of PCV3 in mummified fetuses, stillborn fetuses with myocarditis, perinatal pigs with encephalitis and/or myocarditis, and growing pigs with periarteritis without detection of other significant pathogens including PCV2 [27]. However, PCV3 has been identified as co-infection in multiple herds experiencing reproductive failure (abortion/stillborn) [7,12,17,28], PDNS [7], myocarditis [8], diarrhea [29], respiratory disease [8,30], and neurologic disease [8,31]. Moreover, PCV3 has been broadly identified in clinically healthy animals leading some to question its clinical relevance in the field [32,33].

Due to the increased detection of PCV3 in the global swine population and its identification as a putative pathogen of swine, PCV3 isolation in pure culture become essential for establishing causation. However, until this study, attempts to isolate PCV3 have been unsuccessful. Here we describe the first isolation of PCV3 from perinatal and reproductive failure cases of PCV3-associated disease in the United States and in vivo characterization of the infection by experimental inoculation of a PCV3 isolate in cesarean-derived, colostrum-deprived (CD/CD) pigs.

## 2. Materials and Methods

### 2.1. Cases Description and Herd History

PCV3 isolation was attempted from three diagnostic cases submitted to the Iowa State University Veterinary Diagnostic Laboratory (ISU-VDL) from three different sites reporting weak-born pigs or elevated stillborns and mummies [27]. Samples collected from these cases were previously tested PCR negative for PCV2 and other pathogens associated to reproductive failure including porcine reproductive and respiratory virus (PRRSV), porcine parvovirus (PPV), and atypical porcine pestivirus (APPV) [27].

Case 1 (58312): Brain, heart, liver and kidney tissues from three 1-day-old pigs from a genetic multiplier herd were used for PCV3 isolation. One-third of the litter had congenital tremors. Littermates were weak and not nursing well. Histologic evaluation revealed lymphocytic myocarditis and lymphocytic perivasculitis perivascular cuffs and gliosis in the cerebrum. PCV3 was detected by real-time PCR (qPCR) in the brain (Cq 7.5) and pooled tissues of heart, liver, and kidney (Cq 14.4). PCV3 was detected by in situ hybridization in the cardiomyocytes, neurons, and smooth muscle of arterioles.

Case 2 (44806): Kidney, lung, and heart tissues were used for PCV3 isolation. Samples originated from two different pooled samples (lung and heart; kidney) of six fetuses. Four of the six fetuses were mummified as evidenced by crown-to-rump lengths of 10, 11, 11.5, and 13 cm compared to 22 and 25 cm of the stillborn fetuses. This fetal group originated from a sow herd with a prolonged herd mummy rate of 5%. The farm also reported some weak-born piglets. Histologic evaluation revealed focal lymphocytic myocarditis in one fetus. PCV3 was detected by qPCR in the pooled lung and heart (Cq 9.4) and kidney (Cq 12.5). PCV3 was detected by in situ hybridization in cardiomyocytes.

Case 3 (27734): Brain, lung, and liver from an 8-day-old piglet selected from a litter born with multiple mummified fetuses was used for viral isolation. The farm was a high-health, newly populated herd reporting periods of higher abortion rates and an increase in mummified fetuses. Histologic evaluation revealed lymphocytic myocarditis. PCV3 was detected by qPCR in brain (Cq 13.3), liver (Cq 19.3), lung (Cq 13.5), and pooled lung and spleen (Cq 16.6). PCV3 was detected by in situ hybridization in cardiomyocytes and the smooth muscle of arteries and arterioles.

### 2.2. Sample Processing for Virus Isolation

Tissues were placed individually or as a pool in sterile plastic bags with 15 mL of Eagle’s minimum essential medium (EMEM) (ATCC, Manassas, VA, USA), and homogenized in a Stomacher blender (Model 80, Seward, Ltd., UK) for 10 min. Thereafter, the tissue homogenate was transferred to a 50 mL sterile conical polypropylene centrifuge tube (Corning^®^, Corning, NY, USA) and centrifuged at 3000× *g* for 20 min at 4 °C. The supernatant was collected and filtered through a 0.20 μm syringe filter (Corning^®^) and used as stock material for virus isolation.

### 2.3. Virus Isolation and Propagation in Cell Culture

Pig (*Sus scrofa*) kidney epithelial cells (PK-15), PCV1 and PCV2 free, were cultured in 25 cm^2^ tissue culture flasks (Corning^®^) using growth medium (EMEM supplemented with heat inactivated 10% fetal bovine serum (ATCC, Manassas, VA, USA), 1% penicillin/streptomycin (Gibco^TM^, Thermo Scientific, Waltham, MA) and 25 μg/mL of gentamicin (Gibco^TM^, Thermo Scientific)) and incubated for 16 h at 37 °C with 5% CO_2_. Once cells reached a confluence of 50–60%, the growth medium was decanted from the flask and the cell monolayer was washed twice with phosphate-buffered saline (PBS) (Gibco^TM^, Thermo Scientific). Cells were inoculated with 1 mL of tissue homogenate supernatant or mock infected with infection medium and incubated for 1.5 h at 37 °C with 5% CO_2_ to allow virus adsorption. Thereafter, 4 mL of infection medium (EMEM, 1% penicillin/streptomycin (Gibco^TM^, Thermo Scientific)) and 25 μg/mL of gentamicin (Gibco^TM^, Thermo Scientific) were added to the flask, and incubated for an additional 70.5 h at 37 °C with 5% CO_2_. Virus was harvested by three −80 °C freeze-thaw cycles followed by removal of cellular debris in 15 mL conical tube centrifuged at 3000× *g* for 20 min at 4 °C. Finally, the supernatant was aliquoted into cryovials and identified as passage 1. Nine serial passages (case 3) and four serial passages (case 1 and 2) of virus isolated in passage 1 were performed and confirmed by qPCR and next generation sequencing.

To establish PCV3 growth curve, 24-well plates (Grainer Bio-One, Monroe, NC, USA) were seeded with PK-15 cells diluted 1:10 in growth medium and incubated at 37 °C with 5% CO_2_. When cell confluence reached 50%–60% (16 h), 6 wells (replicates) were inoculated with 220 μL of virus stock (PCV3 ISU27734; passage 3; Cq 17.8; 3.33 × 10^8^ genomic copies per mL) and 6 wells were inoculated with 220 μL of infection medium. Then, plates were incubated for 1.5 h at 37 °C with 5% CO_2_, followed by the addition of 800 μL of infection medium per well, and incubated at 37 °C with 5% CO_2_ for 72 h. The cells were collected every 12 h by three cycles of freezing and thawing (one plate per time point). Plates corresponding to different time points post-inoculation were processed simultaneously, and the genomic copies per mL were assessed by qPCR.

### 2.4. Real-Time PCR (qPCR)

Real-time PCR was used to demonstrate and monitoring PCV3 replication in vitro (cell culture) and in vivo (viremia). Viral DNA extractions from virus culture supernatant, serum, nasal swabs and tissues were performed using the MagMAX-96 Pathogen RNA/DNA kit (Applied Biosystem™, Waltham, MA, USA) with KingFisher™ Flex 96 Deep-Well Magnetic Particle Processor (Thermo Fisher Scientific) following the manufacturer’s instructions. DNA extracts were used to detect the conserved region of the open reading frame 2 (ORF2 or capsid gene) of PCV3 and a highly conserved region of the ORF1 of PCV, to rule out PCV1/PCV2 contamination, using TaqMan™ Fast Virus 1-step Master Mix (Thermo Fisher) with specific primers previously described [7,34]. Internal control RNA (Xeno™, Applied Biosystem^TM^) was included in the master mix to monitor PCR amplification and detection of PCR inhibition. Two positive extraction controls, one negative extraction control, and a negative amplification control were included. qPCR was performed on two different instruments (7500 Fast Real-Time PCR System^TM^, Applied Biosystems; BioRad CFX-96, Hercules, CA, USA) with the following cycling conditions: one cycle at 50 °C for 5 min, one cycle at 95 °C for 20 min, 40 cycles at 95 °C for 3 s, and 60 °C for 30 s. Samples with Cq < 37.0 was considered positive. In some cases, the PCV3 genome copy number per mL was calculated from standard curve created by the ten-fold serial dilution of plasmid DNA containing full genome of PCV3 isolate 29160 (GenBank accession number KT869077).

### 2.5. In Situ Hybridization (RNAscope)

RNAscope technology is a novel in situ hybridization (ISH) assay that was used in this study for detection of viral RNA within intact cells and, therefore, for further demonstration of virus replication both in vitro (culture) and in vivo (tissues). Probes were designed based on the capsid genes 270–1357 region for PCV2 (GenBank accession number KX298474.1) to rule out contamination or coinfection, and 2–1049 region for specific detection of PCV3 (GenBank accession number HQ839721.1). The actual sequence used to design the probe is intellectual property of the provider (Advanced Cell Diagnostics, Newark, CA, USA), and therefore, is not available for public access. PK-15 cells were inoculated in chamber slides (Nunc^®^, Thermo Fisher Scientific), and infected with 200 μL of virus (PCV3 ISU27734; passage 3; Cq 17.8; 3.33 × 10^8^ genomic copies per mL) using the conditions described above. After 72 h post-inoculation cell monolayers were fixed on 10% Neutral Buffered Formalin (NBF). Rehydrated sections were processed (RNAscope 2.5 HD duplex detection kit, Advanced Cell Diagnostics), according to the manufacturer’s instructions. Briefly, cell smears were hybridized with PCV2 and PCV3-specific probes at 40 °C for 2 h in a humidified tray. After a wash step of 2 min in wash buffer, a sequence of amplifiers (amp) was added as follows: amp 1 for 30 min at 40 °C; amp 2 for 15 min at 40 °C; amp 3 for 30 min at 40 °C; amp 4 for 15 min at 40 °C; amp 5 for 30 min at room temperature (RT); amp 6 for 15 min at RT. After amp 6, the red signals (PCV3, peroxidase label) were detected by incubating the slides with a freshly prepared red solution, for 10 min, also at RT. Then, the procedure continued with addition of amp 7 for 15 min at 40 °C; amp 8 for 30 min at 40 °C; amp 9 for 30 min at RT; and amp 10 for 15 min also at RT. Finally, green signals (PCV2, alkaline phosphatase label) were detected by applying a freshly prepared green solution for 10 min at RT. Slides were counterstained with hematoxylin. A wash step of 2 min was performed between amplifiers and/or colorimetric solutions by immersing the slides in the kit wash buffer, with occasional agitation.

### 2.6. Generation of Polyclonal Antibody

Anti-PCV3 polyclonal antibodies (pAbs) were generated to demonstrate PCV3 replication in vitro by immunofluorescence microscopy. BALB/c mice were injected intraperitoneally with PCV3 concentrated by ultracentrifugation from pure cultures. All procedures were approved by the Institutional Animal Care and Use Committee of Iowa State University (log number: 1-3-5381-LM, approval date: June 22, 2016). In brief, 40 mL of PCV3 culture was centrifuged (Optima XPN-100 Ultracentrifuge, Beckman Coulter, Brea, CA, USA) at 91,145× *g* for 3 h at 4 °C, washed twice with PBS to remove culture medium components, resuspended with 40 mL of PBS and centrifuged at 142,414× *g* for 2 h at 4 °C. The final virus pellet was resuspended in 1 mL PBS, mixed with incomplete Freund’s adjuvant at 1:1 volume ratio and used to inoculate intraperitoneally (50 μL per mice) two BALB/c mice, followed by a booster injection two weeks after prime inoculation. Blood serum was harvested and the presence of PCV3 antibodies was evaluated pre- vs. post-inoculation by a PCV3 whole virus-based ELISA described herein. When PCV3 antibody levels reached optimal levels, mice were treated intraperitoneally with Pristane (Sigma-Aldrich, St. Louis, MO, USA) and three days later given an intraperitoneal injection of SP2/O cells (approximately 5 to 8 × 10^6^ cells) to produce ascites fluid containing anti-PCV3 antibodies. After 10 days, the two mice were euthanized, and the fluid was collected and stored at −80 °C.

### 2.7. Indirect Enzyme-Linked Immunosorbent Assays (iELISAs)

PCV3 isolate PCV3/USA/MO/ISU27734/2018 was used to generate a whole virus (wv) indirect ELISA (PCV3 wv iELISA) to evaluate the reactivity of anti-PCV3 polyclonal antibodies developed in mice. A total of 200 mL of PCV3 culture (PCV3 ISU27734; passage 3; Cq 17.8; 3.33 × 10^8^ genomic copies per mL) was centrifuged at 3000× *g* at 4 °C for 20 min to remove cell debris. The virus was concentrated by ultracentrifugation at 142,414× *g* at 4 °C for 2 h, and re-suspended in 1 mL (1:200 of the original culture volume) of PBS pH 7.4 and stored at −80 °C. Following ELISA titration, an optimal dilution of viral pellet (1:200; PBS pH 7.4) was used to coat a 96-well (100 µL per well) polystyrene ELISA plates (Nunc^®^, Thermo Fisher Scientific) and incubated at 4°C for 16 h. The plates were washed five times with PBS-T (0.1% Tween 20), blocked (300 µL/well) with 1% (*w*/*v*) bovine serum albumin blocking solution (BSA) (Jackson ImmunoResearch Inc., West Grove, PA, USA), incubated at 20–22 °C for 3 h, and dried at 37 °C for 4 h.

A PCV3 ORF2 recombinant protein based iELISA (PCV3 rORF2 iELISA) was also developed and used in this study to evaluate the kinetics of the humoral antibody (IgG) response of CD/CD pigs infected with a PCV3 isolate under experimental conditions. A codon optimized gene codifying for PCV3 ORF2 truncated protein (164 amino acids) without the nuclear localization signal (NLS) was synthetized in vitro (Shanghai Genery Biotech Co., Ltd., Shanghai, China). Amplicon was cloned into pCold II expression plasmid and transformed into an *Escherichia coli* BL21 (DE3) host strain according to manufacturer’s instructions (Invitrogen, Carlsbad, CA, USA). The transformants were resuspended and grown in 1 L of Luria-Bertani (LB) medium (Invitrogen) containing 100 µg/mL ampicillin at 16 °C with shaking at 250 rpm. When an A_600_ value of 0.6–0.8 was reached, 0.3 mM IPTG (isopropyl-β-d-thiogalactopyranoside) was added, and cultures were grown for an additional 16 h at 20 °C. Cells were chilled at 4 °C and harvested by centrifugation at 3500× *g* for 15 min. Protein solubility analysis by dodecyl sulfate-polyacrylamide gel electrophoresis (SDS-PAGE) showed that the histidine (His) tag-fused ORF2 PCV3 protein was expressed insoluble as inclusion bodies. Therefore, pellet was resuspended in 40 mL of lysis buffer (20 mM Tris, 300 mM NaCl, pH 8.0), disrupted by ultrasonication (Φ 3, 15%, 3 s on/6 s off, 10 min) (Vibra-Cell sonicator; Sonics & Materials, Newtown, CT, USA) and centrifuged at 15,000× *g* for 30 min at 4 °C. The supernatant was collected and mixed with 2 mL of Ni Sepharose^®^ 6Fast Flow (GE Healthcare, Pittsburgh, PA, USA) and incubated for 1 h 4 °C. Beads were washed with “Buffer C” (20 mM Tris, 300 mM NaCl, 8 M Urea, 20 mM Imidazole, pH 8.0) and the recombinant His tag-fused ORF2 protein (452 aa) was eluted with “Buffer E” (20 mM Tris, 300 mM NaCl, 8 M Urea, 500 mM Imidazole, pH 8.0) according to the manufacturer’s instructions. Finally, PCV3 ORF2 recombinant protein (0.3 mg/mL; Bradford) was dialyzed against 20 mM Tris, 300 mM NaCl, 0.5% SKL, pH 8.0 for coating on to ELISA plates (Appendix A).

For ELISA testing, either PCV3 wv- or recombinant ORF2-coated plate wells were loaded with 100 µL of mice sera (2-fold dilutions starting at 1:100) or pig sera (1:50), incubated at 37 °C for 1 h, and washed 5 times with PBS-T (0.1% Tween 20). Then, 100 μL of either a Peroxidase (HRP)-conjugated AffiniPure Goat Anti-Mouse IgG Fcγ Fragment-specific (subclasses 1+2a+2b+3) (Jackson ImmunoResearch Inc.) diluted 1:20,000 for mice serum, HRP-conjugated goat anti-pig IgG (Fc) diluted 1:10,000 or IgM diluted 1:3000 (Bethyl Laboratories Inc., Montgomery, TX, USA) for pig serum, was added to each well and the plates were incubated at 37 °C for 1 h. After a washing step, the reaction was visualized by adding 100 μL of tetramethylbenzidine-hydrogen peroxide (TMB, Surmodics IVD, Inc., Eden Prairie, MN, USA) substrate solution to each well, and incubate at 20–22 °C for 5 (pig serum) to 10 min (mice serum). The reaction was stopped by adding 100 μL of stop solution (Surmodics IVD, Inc.) to each well. The absorbance was measured at 450 nm using an automated plate reader (Molecular Devices, Sunnyvale, CA, USA). Serum antibody responses in pig and mice were expressed as sample-to-positive (S/P) ratios.

### 2.8. Indirect Immunofluorescence Assay (iIFA)

Mouse polyclonal sera were incubated with PCV3-infected and non-infected cells and visualized by immunofluorescence to demonstrate productive infection. PK-15 cells were seeded in a 96-well flat clear bottom black polystyrene surface treated microplate (CellBIND^®^; Corning^®^), with a concentration of 4 × 10^4^ cells per well using growth medium and incubated at 37 °C with 5% CO_2_. After 16 h (50%–60% cell confluence) cells were inoculated with 25 μL of virus stock (PCV3 ISU27734; passage 6; Cq 24.1; 6.7 × 10^10^ genomic copies per mL) and incubated for 1.5 h at 37 °C with 5% CO_2_, followed by addition of 80 μL of infection medium per well and incubated at 37 °C with 5% CO_2_ for 72 h. After 72 h, cells were fixed with 50 μL/well of 2.5% glutaraldehyde in 0.1 M cacodylate buffer for 20 min at room temperature. Plate was washed three times with PBS. Cells were permeabilized with PBS containing 0.5% Tween 20 for 10 min, followed by three washes with PBS. Polyclonal antibodies obtained from two mice named as “L and N”, and serum from pre-immunized mice were serial diluted in PBS with 0.1% BSA (Jackson ImmunoResearch), pH 7.4, from dilution 1:20 to 1:2560. Serially diluted mice sera were incubated in triplicate 1 h at 37 °C in testing plates. Then, after a 3-times washing step (200 μL of PBS per well and wash), 50 μL of fluorescein isothiocyanate (FITC)-conjugated goat anti-mouse IgG (KPL, Milford, MA, USA) diluted 1:100 (PBS with 0.1% BSA) was added to each well, and 96-well plates were incubated for 45 min at 37 °C and washed as described above. Finally, a DAPI nuclear/DNA counterstaining was performed by addition of 50 μL of Hoechst 33342 (Invitrogen, Carlsbad, CA, USA) diluted 1:500 in PBS pH 7.4 per well, incubated for 10 min at room temperature (19–22 °C) followed by another 3-times washing step. Green fluorescent intensity, indicative of virus replication, was read and quantified in individual wells using a SpectraMax i3x^®^ image cytometer operated with the SoftMax Pro 6.5 software (Molecular Devices, Sunnyvale, CA, USA). The image acquisition settings for the SpectraMax i3x^®^ were as follow: four sites per well (17.28 mm^2^), wavelength 541 nm, 30 ms exposure, and 0 μm focus adjustment. Wells were also observed under an IX83 Inverted Microscope (Olympus Life Science, Waltham, MA, USA) using the fluorescence DAPI and GDF fluorescent channels.

### 2.9. Virus Genome Extraction and Sequencing

Virus recovery after serial passage and virus culture purity was determined by next generation sequencing. Total nucleic acid was extracted from virus culture supernatants using MagMAX Pathogen RNA/DNA Kit (Thermo Scientific) [35]. Double stranded DNA was synthesized using NEXTflex™ Rapid RNA-Seq Kit (Bioo Scientific Corp, TX, USA). The sequencing library was prepared using Nextera XT DNA library preparation kit (Illumina, San Diego, CA, USA) with dual indexing. The pooled libraries were sequenced on an Illumina MiSeq platform using the 300-Cycle v2 Reagent Kit (Illumina) by following standard Illumina instructions.

### 2.10. PCV3 Genome Assembly, Bioinformatics and Phylogenetic Analysis

Raw sequencing reads were pre-processed using Trimmomatic v0.36 to remove adapters and trim low-quality ends [36]. Raw reads and pre-processed reads were subjected to sequencing quality analysis with FastQC (https://www.bioinformatics.babraham.ac.uk/projects/fastqc/). Cleaned reads were then fed to a comprehensive reference-assisted virus genome assembly pipeline [35,37] with modifications. Briefly, publicly available whole genome sequences of the PCV3 were downloaded from NCBI to serve as a collection of reference sequences on the date of July 31, 2018. Quality-trimmed reads were mapped against the reference sequences by using BWA-MEM [38]. Mapped reads were extracted by SAMtools [39] and seqtk (https://github.com/lh3/seqtk). De novo assembly was performed using ABySS v1.3.9 [40]. The resulted contigs were manually checked and trimmed in SeqMan Pro (DNASTAR^®^ Lasergene 11 Core Suite). Gaps were closed by using the primers and protocols as previously reported [7].

For phylogenetic analysis, full genome sequences of 46 representative PCV3 strains from the United States (*n* = 5) rest of America (*n* = 12), Europe (*n* = 5) and Asia (*n* = 23) were obtained from GenBank (https://www.ncbi.nlm.nih.gov/nucleotide/) (acquisition date 22 October 2019).

PCV3 reference strains corresponded to different clades were selected from US strains with the exception of PCV3b due to the absence of US sequences corresponded to this specific clade (PCV3a: PCV3USA/MO/2015 (KX778720), PCV3/USA/29160/ NC/2016 (NC031753), PCV3/USA/MN20162016 (KX898030); PCV3b: PCV3/CN/Jiangxi-B1/2017 (MF589107); and PCV3c: PCV3/USA/SD2016/2016 (KX966193)). Multiple sequence alignment and sequence comparisons were carried out using the ClustalW algorithm by Geneious R9 software (Biomatters Ltd., Auckland, NZ). Phylogenetic analysis was completed using PAUP (phylogenetic analysis using parsimony) version 4.0b10 by the maximum parsimony (MP) method. The MP tree was executed using a heuristic search with stepwise option. The clade of trees was estimated with 1000 replicates of bootstrap (BT) analysis. The trees were visualized with Geneious R9 software based on full sequence of the ORF2. All sequences were translated and PCV3 clustering was evaluated based on the amino acid composition of the ORF2 as previously proposed [39].

### 2.11. Experimental Inoculation of CD/CD Pigs

A pig experimental inoculation study was performed to evaluate the ability of PCV3 isolate ISU27734 of replicating in vivo. The study was approved by AMVC WeSearch DBA VRI (Audubon, IA, USA) Animal Use and Care Committee (BI-S-18-1248). Adequate floor and feeder space was provided in accordance with acceptable animal husbandry practices. Specifically, fourteen 6-week-old CD/CD pigs were individually identified and randomly assigned to two treatment groups. Negative controls (*n* = 6) and PCV3-whole virus inoculated animals (*n* = 8) received 2 mL (1 mL intranasal and 1 mL intramuscular) of infection medium or 2 mL of PCV3 isolate ISU27734 (P6) containing 6.6 × 10^10^ genomic copies per mL (Cq 23.58), respectively. All animals received a subcutaneous injection of an immunostimulant (keyhold limpet hemocyanin emulsified in incomplete Freund’s adjuvant; KLH/IFA) 2 days prior to (right rear leg; 2 mL) and 2 days after inoculation (left rear leg; 2 mL). KLH/IFA contained 1 mg KLH per 1 mL adjuvanted with 1 mL of IFA.

Serum was collected at 0, 7, 14, 21, and 28 days post-inoculation (dpi). All animals were evaluated daily for the presence of respiratory, neurologic, and enteric clinical signs as well as body condition. Animals were euthanized at 28 dpi (10 weeks of age). A set of tissues including lung, heart, liver, spleen, kidneys, small and large intestine, superficial and visceral lymph nodes, tonsil and brain were fixed in 10% neutral buffered formalin, processed by standard technique and stained by hematoxylin and eosin (H&E) technique. Histologic evaluation was performed by a diagnostic pathologist.

### 2.12. Statistical Analysis

A Generalized Linear Model was used to analyze the association between iIFA (fluorescent intensity) results and treatment (pAb) and dilution, with treatment and dilution being the fixed effects, and the three replicates being the random term. Comparisons between cell control and mouse hyperimmune sera, comparisons between mice pre-immune serum vs. hyperimmune sera, and comparison between hyperimmune sera collected from two mice were conducted using Bonferroni adjustment for multiple comparison. For the pig study, one-way ANOVA with Tukey’s correction was used for multiple comparisons with alpha = 0.05. A statistical analysis was performed using SAS version 9.4 (SAS Institute, Inc., Cary, NC, USA,). Data were plotted using GraphPad Prism 8 (GraphPad Software, Inc., San Diego, CA, USA).

### 2.13. Data Availability

New sequences were deposited in GenBank under accession numbers MK058528 (case 3; passage 6), MK058529 (case 3; passage 9), MK568470 (case 1; passage 3), and MK568469 (case 2; passage 3).

## 3. Results

### 3.1. PCV3 Was Successfully Isolated from Perinatal and Reproductive Cases

Virus isolation was attempted on PCV3 qPCR positive porcine samples from perinatal and reproductive cases of PCV3-associated disease with Cq values ranging between 7.5 and 19.3 (Table 1). Three PCV3 strains, PCV3/USA/NC/ISU58312/2018, PCV3/USA/IA/ISU44806/2018, and PCV3/USA/MO/ISU27734/2018, were successfully isolated from three different cases and tissue samples in PK-15 cells. Contamination or coinfection with either PCV1 or PCV2 was ruled out by testing each culture passage by PCV differential PCR.

### 3.2. PCV3 Replicates in PK-15 Cells

After isolation was confirmed, replication of the isolates was confirmed by kinetic of infection in PK-15 cells, detection of expression of viral proteins by immunofluorescence and genome replication in cells by in situ hybridization.

Growth kinetics of PCV3 isolate ISU27734 was evaluated in PK-15 cells (Figure 1). PCV3 DNA detected by qPCR in PCV3-infected cells (6 replicates per time point) increased from 3.61 × 10^8^ at 12 h post-inoculation (hpi), peaked at 60 hpi (5.16 × 10^8^ genomic copies per mL), and slightly declining at 72 hpi (4.23 × 10^8^ genomic copies per mL). Inoculated cells did not demonstrate a distinct cytopathic effect compared to mock-infected PK-15 cells (data not showed).

Virus replication in PK-15 cells was demonstrated by iIFA using two pAbs directed against PCV3 whole virus particles (Figure 2). The reactivity of the mouse hyperimmune (L and N) vs. pre-immune sera was first evaluated at different dilutions by ELISA (Figure 2A) and fluorescent image cytometry (Figure 2B) with no significant differences (*p* > 0.01) in immunoreactivity between pAbs. Further analysis of the fluorescent intensity captured by image cytometry showed that PCV3 pAbs used at 1:20 or 1:40 (no significant difference) as primary antibody provided the highest discrimination between the fluorescent intensity indicative of virus replication and the unspecific fluorescent intensity (background) from the negative control wells (Figure 2B). Absence of PCV3-specific immunofluorescence was reported in PCV3 infected cells incubated with pre-immune mice serum (Figure 2C). Immunofluorescence staining using either mouse PCV3-specific hyperimmune serum “L” (Figure 2D) or “N” (Figure 2E) as primary antibody revealed nuclear and/or intracytoplasmic green immunofluorescence specific for viral antigen detection and, therefore, virus replication.

In addition, virus replication in PK-15 cells was confirmed by in situ hybridization. After 72 hpi, epithelial cells showed strong intracytoplasmic hybridization signals characterized by active-replicating PCV3 (Figure 3).

PCV3 isolate ISU27734 was passaged in PK-15 cells, and the genome sequences of passage 1, 3, 6 and 9 were determined. The analysis revealed 100% genome identity amongst four isolates from all passages evaluated, indicating that the PCV3 isolate ISU27734 is stable at least during 9 passages in cell culture.

### 3.3. Full Length Sequence and Assembly

Eight PCV3 complete genomes were assembled from the three PCV3 isolates of different passages or origin (Table 1), including 4 identical sequences for isolate PCV3/USA/MO/ISU27734/2018, two identical sequences for isolate PCV3/USA/IA/ISU44806/2018, and two identical sequences for isolate PCV3/USA/NC/ISU58312/2018. The genome sequences for isolate ISU27734 (case 3) were deposited to GenBank under the accession numbers of MK058528 and MK058529, for passage 6 and passage 9, respectively. The passages 1 and 3 were not submitted to GenBank due to their 100% identity with passages 6 and 9. The representative genome sequences of isolate ISU44806 (case 2; passage 3) and ISU58312 (case 1; passage 3) were deposited in GenBank under the accession numbers of MK568470 and MK568469, respectively. All sequences were 2000 nt in length with two major ORFs encoding the replication-associated protein (Rep) and the capsid protein (Cap). Comparison between the PCV3 genome sequences reported in the present study, PCV3 isolates ISU44806 and ISU58312 had a pairwise 99.1% nucleotide identity; isolates ISU44806 and ISU58312 showed 99.2% and 98.9% with isolate ISU27734, respectively.

### 3.4. Phylogenetic Analysis Based on PCV3 Complete Genome Sequences

The phylogenetic tree based on PCV3 complete genome sequence, including four representative complete sequences from three PCV3 isolates assembled in this study and 46 representative sequences reported from America, Europe and Asia, are shown in Figure 4. Based on the whole genome structure, the identity on the nucleotide sequence amongst the four isolates varied from 98.75% to 99.95%. The cladogram showed that strain MK568469.1/USA/2018 clustered with the PCV3a2 reference strain NC031753.1/USA/2018 showing 99.5% identity; while both isolates (MK058528 and MK058529 in the subclade PCV3a3) clustered with each other. Finally, the PCV3 isolate included in the PCV3c clade (MK568469.1/USA/2018) clustered with a reference strain MF162298.1/Italy/2017 with a 99.9% nucleotide identity. The identity on the nucleotide sequence of the four isolates compared with previously reported sequence from US varied from 97.9% to 99.8%. While the identity with the sequences reported from the rest of America varied from 98.65% to 99.99%. The identity on the sequence of the four isolates compared with references strains from Europe varied from 98.6% to 99.9% while the nucleotide identity with Asia strains varied from 98.75% to 99.85%. The highest nucleotide identity on the sequence at the whole genome level (99.9%) was observed specifically with isolate MK568470 and one strain from Brazil (MK645718.1/Brazil/2018) and one from Italy (MF162298.1/Italy/2017)

### 3.5. Phylogenetic Analysis Based on PCV3 ORF2 Sequences

The phylogenic analysis of the ORF2, including four sequences of the PCV3 strains isolated in this study and 46 reference sequences reported from America, Europe and Asia, are showed in Figure 5. Based on the phylogenetically distance 3 of the isolates (MK058528 and MK058529, and MK568469) belong to the clade PCV3a and one (MK568470) to the PCV3c. Isolates in the PCV3a clade were further classified in two different subclades: PCV3a1 isolates MK058528 and MK058529 and PCV3a2 isolate MK568469, based on reference sequences reported previously [27,41].

Based on the ORF2 sequence, isolate MK568470 showed a 98.6% nucleotide identity with the other three isolates, while isolate MK568469 showed a 98.1% nucleotide identity with MK058528 and MK058529. No differences in the ORF2 sequence was observed between both isolates included in the PCV3a3 subclade (MK058528 and MK058529). The identity of the four sequences of this study compared to previously reported sequence from the United States varied from 97.9% to 99.8%. However, the identity with the sequences reported from the rest of America varied from 98.1% to 100%, with special emphasis of isolate MK568470 with two strains from Brazil with 100% identity (MK645719.1/Brazil/2018 and MK645718.1/Brazil/2018). The identity with six European PCV3 strains varied from 97.9 % to 100% being the highest nucleotide identity between isolates MK568470 and MF162298.1/Italy/2017. Finally, 23 sequences from different countries in Asia were included, and the nucleotide identity of the ORF2 sequence varied from 97.9 % to 100%, with only 100% identity observed between isolates MK568470 and KY996341.1/South Korea/2017.

In addition, genetic differences resulted in mutations in amino acids 24 and 27 of the Cap protein, i.e., MK058528.1 MK058529, MK568469.1 (A24: R27) and MK568470.1 (V24:K27). Further changes on the amino acid sequences were observed in the isolated MK568469.1 (S 77 T and I 150 L) (Appendix A). These results further validate the genotyping results that were obtained based on the phylogenetic structure.

### 3.6. PCV3 Isolate Replicates in CD/CD Pigs

PCV3 isolate ISU27734 efficiently replicates in CD/CD pigs (*n* = 8) under experimental conditions. Viral loads detected by qPCR (mean log genomic copies per mL) and IgG/IgM antibody responses (mean S/*p* values) detected by PCV3 rORF2 iELISA over the course of the infection are presented in Figure 6. Viremia was first observed at 7 dpi in a single animal (1/8) and it was present in all animals (8/8) inoculated with the virus isolate by 28 dpi. The mean log10 genomic copies per mL at 28 dpi was 6.09 (range 5.70 to 6.93). PCV3 IgM antibody response trend was detected between 7–14 dpi in pigs (5/8) experimentally inoculated with PCV3 isolate ISU27734; however, it was no statistically significant (*p* > 0.05) compared to the negative control group. Contrary, no IgG seroconversion was detected in PCV3 inoculated pigs throughout the study. PCV3 DNA or specific seroconversion was not detected in any pig within the negative control group throughout the study.

Although all animals infected with the PCV3 isolate remained clinically unremarkable throughout the study (data not shown), histological evaluation showed lesions of ongoing multisystemic inflammation in four out of eight animals. There was lymphoplasmacytic myocarditis and periarteritis (Figure 7A,B). PCV3 replication was confirmed by in situ hybridization within myocardiocytes, the tunica media and endothelial cells of arteries, and inflammatory cells (Figure 7C,D). Sections of kidney demonstrate lymphoplasmacytic interstitial nephritis and periarteritis (Figure 8A,B). The presence of hybridization signal confirmed PCV3 replication within inflammatory cells, tubular renal epithelium, endothelial cells and tunica media of arteries (Figure 8C–E). In addition, there was lymphoplasmacytic periarteritis and arteritis of the intestinal serosa (data not shown).

## 4. Discussion

PCV3 has recently been identified as a putative pathogen in the U.S. swine herd with a subset of infections resulting in fetal wastage and multisystemic inflammation [7,8,27]. Although recently discovered, PCV3 DNA has been detected by qPCR in retrospective samples which indicates that the virus was likely circulating in swine population’s worldwide decades prior to the initial reports [15,23,24,25]. While testing is limited to date, it is hypothesized that as testing increases, PCV3 will be identified in more countries and in older samples.

PCV3 is genetically distinct from PCV1 and PCV2 and its inclusion in the genus circovirus was based on virus features typical of circoviruses including similar genomic organization, with two ambisense ORFs flanking the origin of replication (Ori) [7]. While the overall nucleotide identity between PCV1 and PCV2 isolates is 68–76% [42,43,44,45], there is only 35% (ORF2) to 55% (ORF1) nucleotide identity between PCV2 and PCV3 [7]. Therefore, according to recent reports, PCV2 vaccines are unlikely to cross-protect against PCV3 due to presumed antigenic differences between these viruses [46].

In this study, PCV3 was isolated from three diagnostic case submissions to the ISU-VDL from three different sites reporting weak-born pigs or elevated stillborn and mummified fetuses. PCV3 was isolated from multiple tissue including lung (cases 1 and 2), kidney (case 1), heart (cases 1 and 2) and brain (case 2) of perinatal pigs with encephalitis and/or myocarditis and stillborn and mummified fetuses. To the author’s knowledge, this is the first description of PCV3 isolation in vitro.

Virus replication in cell culture was demonstrated by qPCR, IFA, and RNAscope. PCV2/PCV3 differential RNAscope represents an improvement on single-molecular in situ hybridization that facilitates specific detection of low copy PCV3 transcripts in PK-15 infected cells, providing strong evidence of productive replication. Furthermore, the expression of PCV3 viral proteins in PK-15 cells infected with PCV3 was revealed by immunofluorescence using PCV3 pAbs. As described for other PCVs, PCV3 replication in vitro was best achieved by inoculation of semi confluent monolayers of PK-15 mitotically active cells, indicating that PCV3 DNA replication may depend on cellular enzymes expressed during S phase growth and that, in the natural replication cycle, replication may start only when cells have passed mitosis [47]. Porcine circoviruses are characterized by the lack of cytopathic effect in cell culture. However, there is no previous information regarding the PCV3 behavior in cell culture. No cytopathic effect was observed in PCV3 infected PK-15 cells compared to non-infected cell control.

The detection of PCV3 from PK-15 cell lysates by qPCR decreased over the course of passages not as the result of virus dilution in absence of productive viral replication, which was further demonstrated by RNAscope and immunofluorescence, but because of its inefficient or self-limiting replication and/or resistance to growth in vitro. Previously, Zhu et al. [48] reported that the PK-15 cell population is heterogeneous with respect to permissibility to the PCV2 infection with only ~20% of the cell population being susceptible to infection [47], and viral titers that never exceed 10^5^ TCID50 titer [49]. Yet, PCV1 seems to replicate more efficiently than PCV2, which, in turn, seems to replicate more efficiently than PCV3 in PK-15 cells [50]. Indeed, Beach et al. [51] demonstrated that Ori and Rep, which are functionally interchangeable between PCV1 and PCV2, are responsible for the differences in replication efficiencies between PCV1 and PCV2. Fenaux et al. [52] also reported that two amino acid mutations in the capsid gene increased the efficiency of the replication of PCV2 in PK-15 cells. The efficiency of virus replication could be enhanced in vitro by supplementing culture medium with interferon (INF) inhibitors [53,54,55]. Indeed, different studies demonstrated IFN-γ- and IL-2-dependent enhancement of PCV2 replication in porcine-derived cell lines including PK-15 [49,56,57]. Moreover, PCV3 replication could be also enhanced by selecting a homogeneous subpopulation of PK15 cell line as previously described for PCV2 [48]. There are certainly many other factors that could help to enhance PCV3 replication in vitro and that should be investigated.

In this study, the full-length genome sequences of four different isolates were determined using metagenomics sequencing. Phylogenetic analysis has been widely used in the characterization of PCV3 sequences from clinical samples in multiple countries [18,19,26,41,58]. The isolates described in this study presented a high identity on the nucleotide sequence compared to PCV3 sequences previously reported. PCV3 isolates described herein belonged to two different PCV3 clades and two different subclades (PCV3a1, PCV3a2 and PCV3c), indicating the diversity of PCV3 strains circulating in the United States. It has been previously proposed that mutations in the amino acids 24 and 27 of the cap protein could be potential molecular markers to define different PCV3 clades [26]. There have been previous reports of reproductive cases in the United States associated to PCV3b and PCV3c clades but not PCV3a. Phylogenetic analysis of the PCV3 ORF2 demonstrated that the PCV3 isolates described herein are represented in clades PCV3a and PCV3c. Thus, this study further supports indicating that amino acid changes in specific Capsid motif (A24 V; R27 K; S 77 T and I 150 L) can be used as molecular markers for phylogenetic classification.

There was no previous information regarding the characterization of the PCV3 infection dynamic in vivo using a PCV3 isolate and CD/CD pigs as experimental challenge model. The main objective of the in vivo study was to demonstrate the ability of the PCV3 isolate ISU27734 to infect and replicate actively in CD/CD pigs. The use of CD/CD pigs allowed to circumvent passively acquired immunity against PCV2 (highly prevalent in commercial farms) and PCV3 (unknown prevalence). Moreover, CD/CD pigs were raised negative to most swine pathogens. As previously described for PCV2, KLH/IFA was used for immune stimulation to allow permissive replication and influence the immune response to the virus [59]. Based on the results collected from this study, CD/CD pigs inoculated with the PCV3 ISU27734 did not develop clinical disease under the specific conditions of this study. PCV3 appeared to cause a viremia (8/8 pigs) and IgM antibody response (4/8 pigs) which is a clear evidence of infection. No significant IgG seroconversion was detected in any pig throughout the 28-days study period. The lack of IgG response observed in the IgM positive pigs could be related to the time for immunoglobulin class switching from IgM to IgG and the relatively short duration of this study (28 days). Previous studies on PCV2 showed that, in absence of PCV2 maternal antibodies, an anti-PCV2 IgM response, developed within the first few weeks after infection, preceded the IgG response in serum accompanied by long-term PCV2 viremia [60,61]. In addition, Opriessnig et al., [62] observed that pigs developing PCVAD remained PCV2 IgG negatives, suggesting that the normal antibody switching process was circumvented. The absence of any PCV3-specific antibody response in 4/8 pigs observed in this study could be due to similar or other factors including viral dose, type and/or dose of the immune stimuli administered, duration of the study, as well as host-related variations.

The presence of PCV3 has been linked with numerous clinical presentations; however, reproductive failure and multisystemic inflammation seem to be the most consistently reported across the current literature [7,8]. Moreover, natural infection seems to cause perivasculitis in grower and finisher pigs [27]. However, the present study did not attempt to reproduce PCV3-associated disease. As with PCV2, the primary causative agent of PCVAD, clinical disease might be difficult to reproduce in pigs experimentally infected with PCV3 alone [63,64,65]. Nevertheless, four out of eight infected CD/CD pigs presented myocarditis and multisystemic perivasculitis. The lesions were less severe than those observed on original field clinical cases submitted to the ISU-VDL. As with PCV2, the primary causative agent of porcine circovirus-associated disease, the severity of lesions and intensity of the immune response to the virus could be affected by immune stimulation, concurrent infections with other swine pathogens, and/or other stress factor contributing to systemic inflammation. Specifically, PCV3 ISU27734 used for experimental inoculation of the CD/CD pigs was originally isolated from a lung sample collected from an 8-day-old piglet presenting lymphocytic myocarditis [27]. Although the pathogenesis of PCV3 infection is still unclear and the multisystemic inflammation could be not only a result of PCV3 infection, effective PCV3 replication within lesions was further confirmed by RNAscope.

Although the clinical relevance of PCV3 remains under debate and much work remains to be done before stablish an infection model for PCV3, this is the first isolation of PCV3 from perinatal and reproductive cases of PCV3-associated disease and the first in vitro and in vivo characterization of PCV3 infection in PK-15 cells and CD/CD pigs, respectively. The successful isolation of PCV3 is a promising step forward to better understand the infection dynamics, pathogenicity and virulence of PCV3.

## Figures and Tables

**Figure 1 viruses-12-00219-f001:**
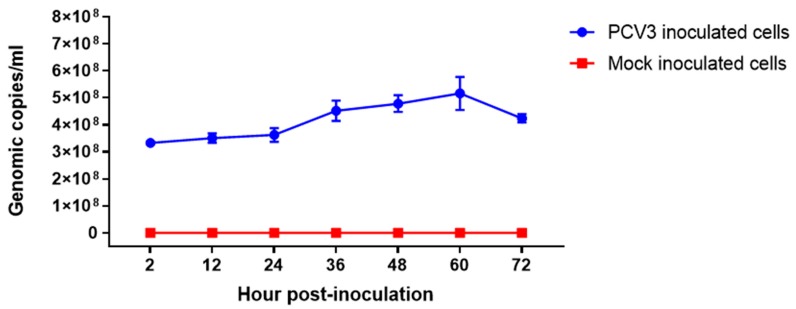
In vitro PCV3 growth kinetic curve. PK-15 cell cultures (six replicates per time point) were inoculated with PCV3 isolate ISU27734 passage 3 containing 3.33 × 10^8^ genomic copies per mL (Cq 17.8) or mock-inoculated (infection medium), and incubated at 37 °C with 5% CO_2_ for 72 h. Time 2 corresponded to cell culture 2 h after virus inoculation. Virus was harvested every 12 h and the genomic copies per mL were assessed by qPCR. Each time point is represented by the mean of six technical replicates and bars represent the standard error (SE) of the mean.

**Figure 2 viruses-12-00219-f002:**
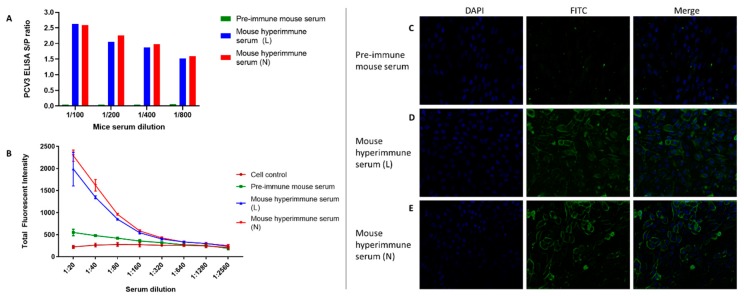
Detection of PCV3 replication in vitro by indirect fluorescent antibody assay (iIFA). PK-15 cells were infected with PCV3 isolate ISU27734 (passage 6). The reactivity PCV3-specific mouse antisera collected from two mice (L and N) was first evaluated and compared to pre-immune sera at different dilutions by ELISA (panel 2**A**) and IFA (panel 2**B**). Subsequently, PCV3 infection of PK-15 cells was demonstrated using PCV3-specific mouse hyperimmune serum diluted 1:20 as primary antibody, FITC-labelled goat anti-Mouse IgG (1:100) was used as secondary antibody and DAPI (1:500) as nuclear counterstain. Absence of cellular reactivity was observed in PCV3 infected cells stained with pre-immune mouse serum (panel 2**C**). Contrary, a strong green intracytoplasmic fluorescent reactivity was observed in PCV3 positive infected cells using PCV3 hyperimmune sera L (panel 2**D**) or N (panel 2**E**) as primary antibody for staining. Images were obtained using a fluorescence inverted microscope (IX83, Olympus) at 40× magnification.

**Figure 3 viruses-12-00219-f003:**
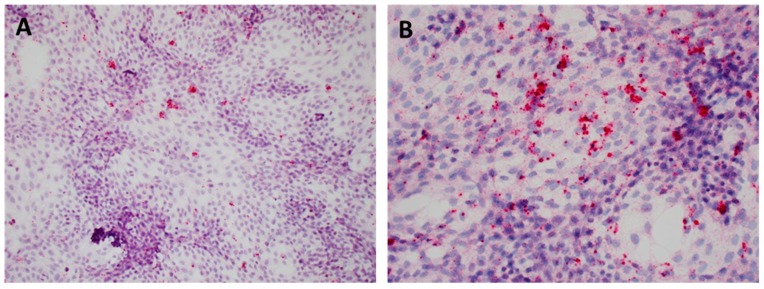
PK-15 infected cells show strong intracytoplasmic hybridization signal. (**A**) An RNA cocktail probe targeting PCV3 ORF2 showed positive signal (red) in individual cells and occasionally clustering cells (20×). (**B**) Positive cells are characterized by small red dots diffusely distributed in the cytoplasm of epithelial cells (40×). Negative cells in purple.

**Figure 4 viruses-12-00219-f004:**
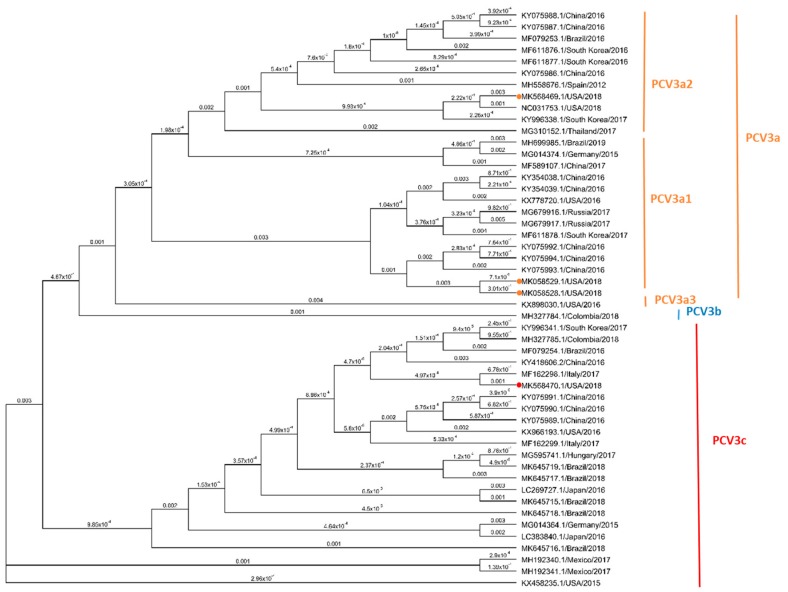
Phylogenetic analysis based on four complete genome sequences of three PCV3 isolated and 46 reference strains. The solid circles indicate the sequences of four viruses isolated in this study. Maximum parsimony (MP) tree based on the nucleotide sequences of the whole genome was protein reconstructed using Geneious R9 software. The value along the branches represent substitutions per site. The clade of trees was estimated using 1000 replicates of bootstrap (BT) analysis. All PCV3 strains were divided into three clades, PCV3a (orange), PCV3b (blue) and PCV3c (red). PCV3a was further divided into three subclades.

**Figure 5 viruses-12-00219-f005:**
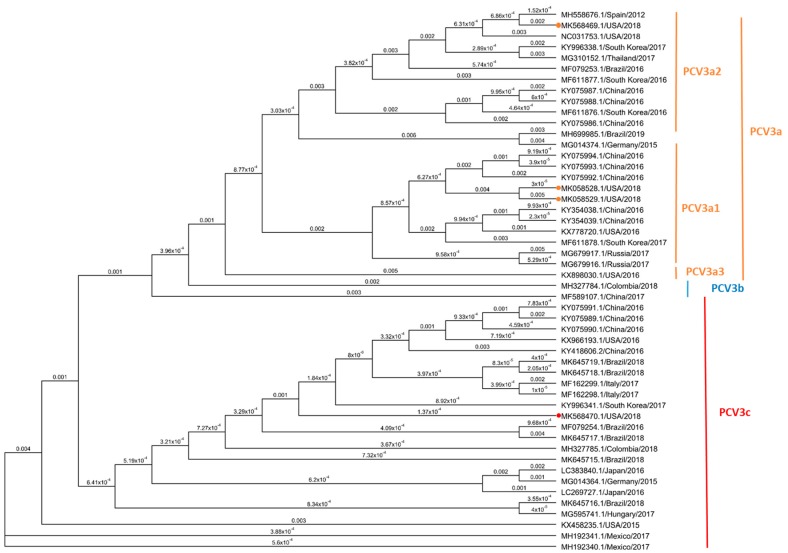
Phylogenetic analysis based on four full-length Cap (ORF2) protein sequences of three PCV3 isolates and 46 reference strains. Solid circles indicate the sequences of four viruses isolated in this study. Maximum parsimony (MP) tree based on the nucleotide sequences of the cap protein reconstructed using Geneious R9 software. The value along the branches represent substitutions per site. The clade of trees was estimated with 1000 replicates of bootstrap (BT) analysis. All PCV3 strains were divided into three clades, PCV3a (orange), PCV3b (blue) and PCV3c (red), based on two amino acids mutation (A24V and R27K) and phylogenetic relationship of cap gene. PCV3a was further divided into three subclades.

**Figure 6 viruses-12-00219-f006:**
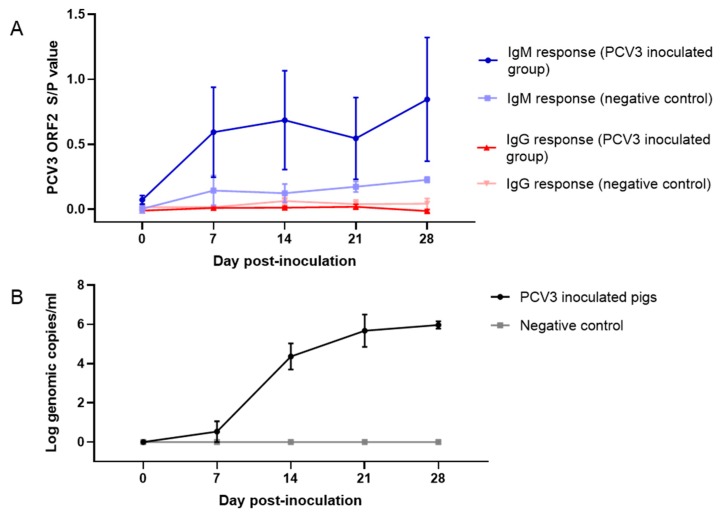
Infection dynamics in 6-week-old CD/CD pigs (*n* = 14) following experimental inoculation with either 2 mL of infection medium (*n* = 6; negative control group) or 2 mL of PCV3 isolate ISU27734 (*n* = 8; PCV3 inoculated group). IgG/IgM antibody responses (mean S/*p* values) detected by PCV3 rORF2 iELISA over the course of the infection (**A**). Viral loads detected by qPCR (mean log genomic copies per mL) over the course of the infection (**B**).

**Figure 7 viruses-12-00219-f007:**
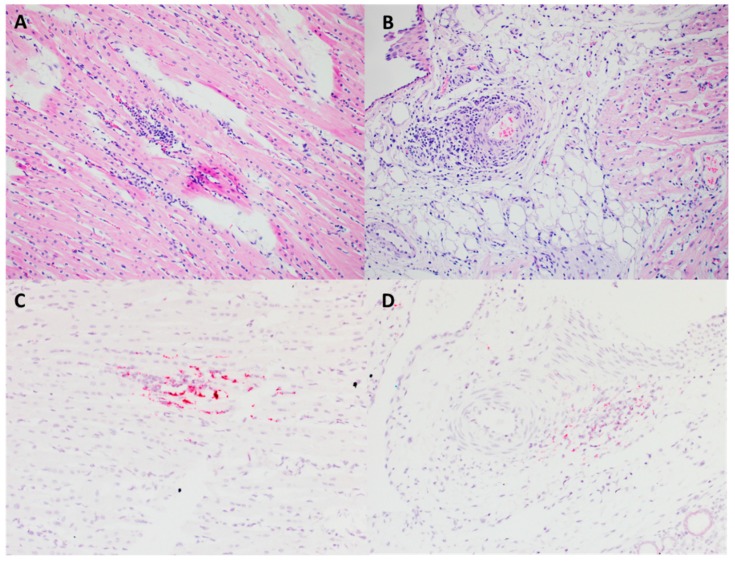
Microscopic lesions and RNAscope in situ hybridization (ISH) on tissues collected at necropsy (28 dpi) from 6-weeks-old CD/CD PCV3 inoculated pigs. The myocardium is multifocal infiltrated by small aggregates of lymphocytes and plasma cells. H&E (20×) (**A**). The tunica adventitia and media of multiple arterioles of the heart are infiltrated by lymphocytes and plasma cells. H&E (20×) (**B**). PCV3 was confirmed by ISH in the cytoplasm of cardiomyocytes and inflammatory and endothelial cells. H&E counterstain (20×) (**C**). Virus replication was detected by ISH within smooth muscle of arteries. H&E counterstain (20×) (**D**).

**Figure 8 viruses-12-00219-f008:**
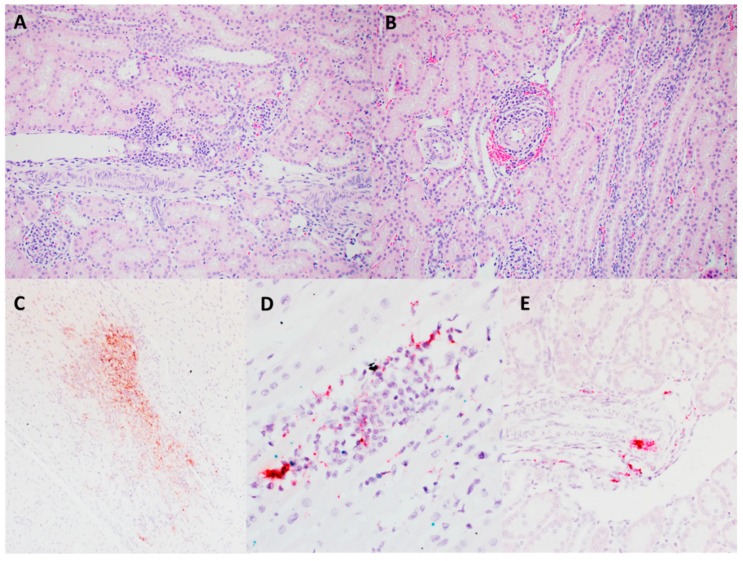
Microscopic lesions and RNAscope in situ hybridization on tissues collected at necropsy (28 dpi) from 6-weeks-old CD/CD PCV3 inoculated pigs. The cortical intestitu is multifocal infiltrated by small aggregates of lymphocytes and plasma cells. H&E (20×) (**A**). The tunica adventitia and media of multiple cortical arterioles are infiltrated by lymphocytes and plasma cells. H&E (20×) (**B**). PCV3 was confirmed by ISH within inflammatory foci (**C**,**D**) endothelial cells (**D**) within smooth muscle of arteries (**E**) H&E counterstain (20×).

**Table 1 viruses-12-00219-t001:** Summary of PCV3 isolation from clinical cases and qPCR values (quantitation cycle or Cq values) during passages in cell culture.

Isolate and Original Tissue	Case	GenBank Accession #	PCV Differential qPCR from Original Tissue			PCV3 qPCR Results ^4^								
			PCV1	PCV2	PCV3	P1	P2	P3	P4	P5	P6	P7	P8	P9
ISU58312 ^1^	1	MK568469												
Heart, liver, kidney homogenate			≥37.0	≥37.0	14.4	19.1	22.3	23.7	27.1 ^5^	NP	NP	NP	NP	NP
Brain			≥37.0	≥37.0	7.5	12.3	15.5	17.5	20.9 ^5^	NP	NP	NP	NP	NP
ISU44806 ^2^	2	MK568470												
Lung, heart homogenate			≥37.0	≥37.0	9.4	14.1	16.8	18.5	21.5 ^5^	NP	NP	NP	NP	NP
Kidney			≥37.0	≥37.0	12.5	16.7	19.6	21.1	24.1 ^5^	NP	NP	NP	NP	NP
ISU27734 ^3^	3	MK058528, MK058429												
Lung			≥37.0	≥37.0	13.5	14.1 ^5^	16.1	17.8 ^5^	19.1	22.2	24.1 ^5^	25.4	27.7	29.9 ^5^
Brain			≥37.0	≥37.0	13.3	19.8	22.9	25.3	28.6	33.5	≥37.0	≥37.0	-	-
Liver			≥37.0	≥37.0	19.3	26.8	30.7	≥37.0	≥37.0	-	-	-	-	-

^1^ PCV3/USA/NC/ISU58312/2018, ^2^ PCV3/USA/IA/ISU44806/2018, ^3^ PCV3/USA/MO/ISU27734/2018, ^4^ Each passage was run by PCV differential qPCR and the result was negative (≥37.0) for PCV1 and PCV2, ^5^ Passage used for next generation sequencing, NP, Passage not performed.

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
