# Peer review of "Isolation of PCV3 from Perinatal and Reproductive Cases of PCV3-Associated Disease and In Vivo Characterization of PCV3 Replication in CD/CD Growing Pigs"

_viruses, 2020, doi:10.3390/v12020219_

Round 1

Reviewer 1 Report

My concern is still in Table 1 and Fig. 1.  Now that authors have indicated in Fig. 1 that their input copy number at 0 hour is 3.33 x 108 copy (which is very high) and the viral replication only double in 72 hours, that means the authors still have not captured the essence of growing this PCV3 in cell culture.  This judgement is further manifested in Table 1 wherein the Cq numbers increase with further passages suggesting a diluting effect of the initial inoculum. No wonder the authors indicate that "not removing the initial inoculum (tissue homogenate)" is critical. Certainly this is not an easy virus to work with.

I accept the data  in Fig. 2, 6, 7, 8 indicating that PCV3 did replicate in experimental pigs.

Author Response

Re: Manuscript ID# viruses-725756

Dear Reviewer,

Thank you for your additional comments. We hope that we have addressed all your concerns. Line numbers listed below refer to the edited manuscript with "markup" option turned on.

Sincerely,

Dr.  Luis G. Giménez-Lirola

Responses to reviewer 1

My concern is still in Table 1 and Fig. 1.  Now that authors have indicated in Fig. 1 that their input copy number at 0 hour is 3.33 x 108 copy (which is very high) and the viral replication only double in 72 hours, which means the authors still have not captured the essence of growing this PCV3 in cell culture.  This judgement is further manifested in Table 1 wherein the Cq numbers increase with further passages suggesting a diluting effect of the initial inoculum. No wonder the authors indicate that "not removing the initial inoculum (tissue homogenate)" is critical. Certainly this is not an easy virus to work with.

Response:  Thank you for your comments. We agree with this reviewer on the difficulty for PVC3 isolation. Certainly, the detection of PCV3 from PK-15 cell lysates by qPCR decreased over the course of passages (Line 568). Evidences from this study as well as difficulties with virus isolation recognized from other groups with broad experience on PCV2 isolation and propagation indicate that PCV3 replicates less efficiently than PCV2 which, in turn, replicates less efficiently than PCV1.

In the original version of the manuscript, we already discussed the fact that PCV3 seems not replicate efficiently in cell culture and that additional efforts towards enhancing PCV3 replication in vitro are necessary. That said, even if it was yet slow and inefficient or self-limiting, we believe that we have brought together enough evidence to demonstrate PCV3 replication in PK-15 cells:

PCV3 ISU27734 isolate used for experimental inoculation of CD/CD pigs was further propagated through passage 9 (it would be impossible to reach this passage in absence of replication). Authors may understand that real-time PCR alone lack spatial resolution and its high analytical sensitivity often makes detection alone unreliable. However, effective or productive PCV3 replication in vitro was further demonstrated by RNAscope and immunofluorescence: PCV2/PCV3 differential RNAscope represents an improvement on single-molecular FISH that facilitates detection of low copy transcripts in PK-15 infected cells (also performed on tissues of pigs experimentally inoculated with PCV3 isolate). Immunofluorescence was used to demonstrate expression of PCV3 proteins in PK-15 cells infected with PCV3.

Nevertheless, the discussion has been extensively reviewed for further clarification.

Line 555-559: “Virus replication in cell culture was demonstrated by qPCR, IFA, and RNAscope. PCV2/PCV3 differential RNAscope represents an improvement on single-molecular in situ hybridization that facilitates specific detection of low copyPCV3 transcripts in PK-15 infected cells, providing strong evidence of productive replication. Furthermore, the expression of PCV3 viral proteins in PK-15 cells infected with PCV3 was revealed by immunofluorescence using PCV3 pAbs”. Line 567-575: “The detection of PCV3 from PK-15 cell lysates by qPCR decreased over the course of passages not as the result of virus dilution in absence of productive viral replication, which was further demonstrated by RNAscope and immunofluorescence, but because of its inefficient or self-limiting replication and/or resistance to growth in vitro. Previously, Zhu et al. [48] reported that the PK-15 cell population is heterogeneous with respect to permissibility to the PCV2 infection with only ~20% of the cell population being susceptible to infection [47], and viral titers that never exceed 105 TCID50 titer [49]. Yet, PCV1 seems to replicate more efficiently than PCV2, which, in turn, seems to replicate more efficiently than PCV3 in PK-15 cells [50]. Line 579-587: “The efficiency of virus replication could be enhanced in vitro by supplementing culture medium with interferon (INF) inhibitors [53-55]. Indeed, different studies demonstrated IFN-γ- and IL-2-dependent enhancement of PCV2 replication in porcine-derived cell lines including PK-15 [49,56,57]. Moreover, PCV3 replication could be also enhanced by selecting a homogeneous subpopulation of PK15 cell line as previously described for PCV2 [48]. There are certainly many other factors that could help to enhance PCV3 replication in vitro and that should be investigated”.

I accept the data in Fig. 2, 6, 7, 8 indicating that PCV3 did replicate in experimental pigs.

Response:  Thank you for your comment.

Reviewer 2 Report

PCV3 is a novel pig virus of unknown importance for health, being detected in both, healthy and clinically affected animals. The major obstacle to study PCV3 pathogenic potential is due to difficulties of PCV3 isolation in vitro. This manuscript is the original and very important contribution in this respect, being the first report of PCV3 isolation that was confirmed in several ways.

The manuscript is generally well written and easy to follow. However, the authors must have been in great hurry as it contains some edits that were not finished before the submission. Also there are some editorial problems, like captions disconnected from their figures.

Specific comments

The authors mix viral load figures given in Cq’s and genomic copies per ml. It should be unified and this reviewer invites the authors to give viral load values in genomic copies instead of Cq’s.

Line 154-155. PCR would certainly be enough to confirm freedom from PCV2 contamination. Or even preferred over in situ hybridization being more sensitive in the detection of minute amount of the virus.

Line 158-159. What was the copy number in the volume of the inoculum used. What was the volume?

Table 1. Please provide copy numbers obtained in qPCR.

Figure 1. The figure has two graphs. And apparently they show the same data. Seems like authors carelessness…

Sections 3.4 and 3.5. The value of providing genomic sequences of new PCV3 strains is obvious but in this manuscript extensive comparison of full genome and ORF2 of these new strains does not improve the quality of the article. Rather opposite, it is distracting. The analysis could be much shortened and only one phylogenetic tree presented (ORF2 perhaps). Please also consider shortening of discussion of this results, lines 570-584.

Line 419 and elsewhere: the term of “nucleotide identity” should be used instead of “homology” when comparing genomic fragments between the different strains of the same species.

The discussion is missing some comments on the interesting results. For example, provide some comparison on IgM and IgG seroconversion observed in PCV2 challenge experiments in CD/CD pigs? Similarities and differences.

Minimal increase in copy numbers in PK15 between 2 and 60 hours post inoculation and rather limited number of infected cells detected in situ may indicate that PCV3 contained in the inoculum was able to infect some cells (very few) and to replicate but not able to spread between the cells. Could the authors discuss it and eventually compare to PCV2? What efforts can be proposed to enhance PCV3 replication (line 566)?

In summary, the most important message from this article is that PCV3 can replicate in PK15 cells and the in vitro isolates are infectious for CD/CD pigs. Moreover, mild myocarditis and perivasculitis was detected, what previously was observed in natural infections. Also, PCV3 seems to need some special stimuli in vivo to replicate efficiently.

Author Response

February 12, 2020

Re: Manuscript ID# viruses-725756

Dear Reviewer,

Thank you for your additional comments. We hope that we have addressed all your concerns. Line numbers listed below refer to the edited manuscript with "markup" option turned on.

Sincerely,

Dr.  Luis G. Giménez-Lirola

Responses to reviewer 2

PCV3 is a novel pig virus of unknown importance for health, being detected in both, healthy and clinically affected animals. The major obstacle to study PCV3 pathogenic potential is due to difficulties of PCV3 isolation in vitro. This manuscript is the original and very important contribution in this respect, being the first report of PCV3 isolation that was confirmed in several ways.

The manuscript is generally well written and easy to follow. However, the authors must have been in great hurry as it contains some edits that were not finished before the submission. Also there are some editorial problems, like captions disconnected from their figures.

Response:  Thank you for your comments and apologize for the confusion that we may have created during previous review/edition. We did modifications on Table 1 and some of the figures. Figure 1 was modified and we originally decided to keep the previous figure so reviewers can see changes. However, we understand that this can be confused; therefore, we have removed the old figure 1. Figures 2-5 remains, only with minor editing in figure captions. Figure 6 is now Supplementary Figure S2. Therefore, previous Figure 7 become Figure 6, previous Figure 8 become Figure 7, and previous Figure 9 become Figure 8. Also, there is a new Supplementary Figure S1 (suggested on previous review) that support the M&M section describing the expression and purification of PCV3 ORF2 protein used to assess antibody response after experimental infection of CD/CD pigs with one of the PCV3 isolates.

Supplementary Figures S1 and S2 and their captions were included in separate attachment.

Specific comments

The authors mix viral load figures given in Cq’s and genomic copies per ml. It should be unified and this reviewer invites the authors to give viral load values in genomic copies instead of Cq’s.

Response: We have now provided genomic copies per ml for those experiments for which we run quantitative PCR, i.e., virus kinetic curve, RNAscope, IFA and animal study. However, we did not run quantitative PCR during virus passage in cell culture. This is why we only presented Cq values in Table 1.

Line 154-155. PCR would certainly be enough to confirm freedom from PCV2 contamination. Or even preferred over in situ hybridization being more sensitive in the detection of minute amount of the virus.

Response:  Thank you for your comment. Actually, in this study PCV1/PCV2 contamination/coinfection was ruled out by using both real time PCR and RNAscope. This has been further clarified in different sections of the manuscript (M&M section 2.4 and 2.5; and Result section Line 342-343).

Line 158-159. What was the copy number in the volume of the inoculum used. What was the volume?

Response: Copy number and volume of the inoculum has been added (Line 161).

Table 1. Please provide copy numbers obtained in qPCR.

Response:  Unfortunately, we did not run quantitative PCR during virus passage in cell culture.

Figure 1. The figure has two graphs. And apparently they show the same data. Seems like authors carelessness…

Response:  Apologize for the confusion we could have caused by keeping both old and new Figure 1. We thought that it would be beneficial to keep both figures so the reviewers could see specific changes. However, we understand that this lead to confusion and; therefore, we have removed previous Figure 1.

Sections 3.4 and 3.5. The value of providing genomic sequences of new PCV3 strains is obvious but in this manuscript extensive comparison of full genome and ORF2 of these new strains does not improve the quality of the article. Rather opposite, it is distracting. The analysis could be much shortened and only one phylogenetic tree presented (ORF2 perhaps). Please also consider shortening of discussion of this results, lines 570-584.

Response: We do of course recognize and respect your opinion on this regards. However, we have to consider that this information may be relevant and valuable to other colleagues more interested on phylogenetic aspects of the virus. Overall, we are seeking the greatest possible impact of this article and, therefore, we find it hard to decide which information is relevant or irrelevant. Nevertheless, the discussion has been reviewed for simplicity (Line 588-602).

Line 419 and elsewhere: the term of “nucleotide identity” should be used instead of “homology” when comparing genomic fragments between the different strains of the same species.

Response:  This has been corrected throughout the manuscript.

The discussion is missing some comments on the interesting results. For example, provide some comparison on IgM and IgG seroconversion observed in PCV2 challenge experiments in CD/CD pigs? Similarities and differences.

Response:  Further discussion relative to seroconversion has been included in the discussion (Line 615-624).

Minimal increase in copy numbers in PK15 between 2 and 60 hours post inoculation and rather limited number of infected cells detected in situ may indicate that PCV3 contained in the inoculum was able to infect some cells (very few) and to replicate but not able to spread between the cells. Could the authors discuss it and eventually compare to PCV2? What efforts can be proposed to enhance PCV3 replication (line 566)?

Response:  The manuscript has been reviewed to introduce some discussion on this regard (Line 579-587).

In summary, the most important message from this article is that PCV3 can replicate in PK15 cells and the in vitro isolates are infectious for CD/CD pigs. Moreover, mild myocarditis and perivasculitis was detected, what previously was observed in natural infections. Also, PCV3 seems to need some special stimuli in vivo to replicate efficiently

Response:  Thank you for your comments.

This manuscript is a resubmission of an earlier submission. The following is a list of the peer review reports and author responses from that submission.

Round 1

Reviewer 1 Report

This manuscript describes for the first time the isolation and characterization of PCV-3 originated from pigs. Virus was isolated on PK-15 cells from organs of three 1 and 8 days piglets and fetuses. Virus was detected by qPCR, in situ hybridization, ELISA developed, and indirect immunofluorescence, depending on samples analyzed. In addition whole genome sequences of viral genome revealed similar organization typical for PCV genomes. Phylogenetic analysis indicated that viral sequences were grouped in several clades indicating genetic variability of viral samples. Experimental infection of CD/CD pigs confirmed viremia and production of IgM antibodies. No clinical signs were observed in infected pigs but virus was confirmed in some organ lesions by histology examination and in situ hybridization.

Despite PCV-3 was identified not only in USA but also in many other countries pathogenesis of this infection is not clear yet. Data presented in manuscript are unique and open the step forward to better study of PCV-3 properties and now we are closer to the preparation of infection model. This work is fundamental work for further development of circovirus research.

Manuscript is carefully written, all experiments are well designated, and results are clearly interpreted with solid conclusions.

Minor points:

No doubt that samples were also analyzed for other viruses. Why not to give this results (for example to M&M) to provide deeper insight on pigs where PCV-3 was detected and isolated? Many other investigators were unsuccessful with cultivation of virus. Maybe it would be useful to give more “hidden” details for the cultivation process. Something is mentioned in Discussion but not in M&M. The virus dose is possible to measure as number of genomes only? Sequences are homologous or non-homologous only (not in percentage). To compare sequences, the percentage of nucleotide identity is used. Please correct throughout manuscript.

Reviewer 2 Report

Authors tried to be the first to isolate PCV3 with subsequent in vitro and in vivo characterization.  Therefore, the title of this manuscript is misleading at first sight.  It sounds like this author get the first isolate of PCV3.  I suggest  change the title.

Overall this is a manuscript difficult to read being drowned in technical details.  As the authors stated, the pathogenesis of PCV3 is still under debate. Plus a great many subtypes of PCV2 as well as many circulating recombinant forms (CRF) of PCV2 are present.  Plus as in PCV2 a variety of non-infection factors influence the validity of the animal model used trying to consistently to reproduce the disease experimentally.  It is certainly not easy for PCV3.

The manuscript is way too long. Certainly I believe the authors have isolated the PCV3.  However, authors did not try to actively ruled out whether their isolates contained PCV2, because in the field, co-infection with PCV2 is not unlikely, the same protocol can be used to isolate PCV2 (therefore possibility of co-isolate).  Further, the same protocol used to purify PCV3, such as ultracentrifugation etc, can also co-purify PCV2.  Authors must rule out   PCV2 in every step of the way, by PCV2 qPCR in their isolate and growth kinetic curve, by PCV2 qPCR and ISH in samples collected from infected pigs since many lesions may be attributed to PCV2 etc.

I would  prefer to see the purpose of  each experiments and the logic of how they interpret the results and how to differentiate, rather than the technical details, which are also important for science.

Specific comments:

line 121 and Fig. 1: the copy number contained in the 220 ul inoculant must be indicated. This is important to interpret the viral replication and growth kinetics.

line 159: what is the purpose of using PCV2 probe? To exclude the possibility of PCV2 co-infection or contamination as addressed above?  You did not explain the logic of using it.

line 163: the polyclonal serum usually do not have the power of resolution.

Line 183: as  indicated above, the same ultracentrifugation can co-purify PCV2.

line 190-211: the cloning and expression of PCV3 ORF2 recombinant protein by itself need more verification (by itself can be  short paper).  If the expressed protein is not characterized well, you cannot expect valid results. 

Line 283: what is special about CD/CD pigs? Why used this pigs? Are they PCV2-free.

p8, Table 1: what are the numbers in qPCR results meant?  They are not explained in the text. or delete it.

p. 9, Fig. 1: the replication between 0-12 hours is very fast.  This is contradictory to what you said in line 528 (slow).  You need to know the copy number of your inoculant to show the rate of virus replication.

line 336: 5.16 x 108 genomic copies is in the peak at 60 hours, not the inoculant at hour 0.

line 393 and line 417: there is slight discrepancy in the results using two methods. You did not discuss it. Choose one of them that you believe is the most reliable, and delete the other one. To choose retaining Fig. 6 can better correlate with data listed in Fig. 6.

Figure 6: This figure is voluminous. You can replace it with a Table listing key mutations.

line 453: "classification" not "calcification".

Discussion: in discussion you compare your model with that of PCV2.  Why not construct a two-column table listing the difference or similarity in every aspect.  It will be a lot clear for the readers at a glance.

Overall the manuscript can be shortened tremendously.

Reviewer 3 Report

The manuscript reports the isolation of PCV3 strains from perinatal and reproductive cases and their characterization in vitro and in vivo.

The subject of this manuscript is extremely important.

Anyway, the work has an important flaw: data concerning virus isolation are contradictory. Indeed, the quantification of PCV3 in PK-15 cell cultures inoculated with positive tissue homogenates, shows a very slight (less then 1log) increment in 72h post-infection. Moreover authors demonstrate a intracytoplasmic fluorescent reactivity in PCV3 positive infected cells using PCV3 hyperimmune sera. On the other hand no increment in viral DNA is demonstrated during passages in cells culture(Table 1), but rather Cq constantly increases of about 2-3 Cq coherently with a viral dilution.

The high viral titre of the initial homogenates explains the possibility of performing a whole genome sequencing after three passages of sub-culture and the efficient infection piglet using the cell culture supernatant.

The study provides other important data, mainly concerning experimental infection, but the ambiguity of the results relative to viral isolation must be resolved, or the discussion of data and the conclusions revised.